# Use of antimicrobials during the COVID-19 pandemic: A qualitative study among stakeholders in Nepal

**Binod Dhungel**[1], **Upendra Thapa Shrestha**[1], **Sanjib Adhikari**[1], **Nabaraj Adhikari**[1], **Alisha Bhattarai**[2], **Sunil Pokharel**[3], **Abhilasha Karkey**[3,4], **Direk Limmathurotsakul**[3,5], **Prakash Ghimire**[1], **Komal Raj Rijal**[1], **Phaik Yeong Cheah**[3,5], **Christopher Pell**[6], **Bipin Adhikari**[3,5]*

1 Central Department of Microbiology, Tribhuvan University, Kathmandu, Nepal, 2 Manmohan Cardiothoracic Vascular and Transplant Center, Institute of Medicine, Tribhuvan University, Kathmandu, Nepal, 3 Center for Tropical Medicine and Global Health, Nuffield Department of Medicine, University of Oxford, Oxford, United Kingdom, 4 Oxford University Clinical Research Unit, Patan Academy of Health Sciences, Lalitpur, Nepal, 5 Mahidol-Oxford Tropical Medical Research Unit, Faculty of tropical Medicine, Mahidol University, Bangkok, Thailand, 6 Amsterdam Institute for Social Science Research, University of Amsterdam, Amsterdam, The Netherlands

* Bipin@tropmedres.ac

**Data Availability Statement:** The data is available upon request to the Mahidol Oxford Tropical Medicine Research Unit Data Access Committee

## Abstract

The COVID-19 pandemic was a major public health threat and the pressure to find curative therapies was tremendous. Particularly in the early critical phase of the pandemic, a lot of empirical treatments, including antimicrobials, were recommended. Drawing on interviews with patients, clinicians and drug dispensers, this article explores the use of antimicrobials for the management of COVID-19 in Nepal. A total of 30 stakeholders (10 clinicians, 10 dispensers and 10 COVID-19 patients) were identified purposively and were approached for an interview. Clinicians and dispensers in three tertiary hospitals in Kathmandu assisted in the recruitment of COVID-19 patients who were undergoing follow-up at an out-patient department. Interviews were audio recorded, translated and transcribed into English, and were analyzed thematically. The respondents report that over-the-counter (OTC) use of antibiotics was widespread during the COVID-19 pandemic in Nepal. This was mostly rooted in patients' attempts to mitigate the potential severity of respiratory illnesses, and the fear of the stigmatization and social isolation linked to being identified as a COVID-19 patient. Patients who visited drug shops and physicians reportedly requested specific medicines including antibiotics. Clinicians reported uncertainty when treating COVID-19 cases that added pressure to prescribe antimicrobials. Respondents from all stakeholder groups recognized the dangers of excessive use of antimicrobials, with some referring to the development of resistance. The COVID-19 pandemic added pressure to prescribe, dispense and overuse antimicrobials, accentuating the pre-existing OTC use of antimicrobials. Infectious disease outbreaks and epidemics warrant special caution regarding the use of antimicrobials and specific policy response.

([http://www.tropmedres.ac/data-sharing-policy-v1-0.pdf](http://www.tropmedres.ac/data-sharing-policy-v1-0.pdf)).

**Funding:** The study was funded by Wellcome Trust [220211, 096527], United Kingdom. For the purpose of Open Access, the author has applied for a CC by public copyright license to any Author Accepted Manuscript version arising from this submission. The funder has no role in study design, data collection and analysis, decision to publish, or preparation of the manuscript.

**Competing interests:** The authors declare that they do have no competing interests. Bipin Adhikari serves as an associate editor at PloS Global Public Health.

## Introduction

The viral pandemic caused by Severe Acute Respiratory Syndrome Corona Virus 2 (SARS-CoV-2) has affected 765 million people, resulting in 6.9 million deaths worldwide as of 5th May 2023 [1]. Because of the efforts made by scientific communities across the globe, effective therapies and vaccines were developed at an unprecedented speed [2, 3]. Nonetheless, diverse variants of the virus are emerging and constantly posing challenges, thereby resulting in decreasing but sustained morbidity and mortality [4].

Concurrently, the world faces the threat of antimicrobial resistance (AMR), which may have even more catastrophic health impacts in the long term. Although its impacts are less obvious now, drug-resistant pathogens are now responsible for approximately 700,000 deaths per year, with this potentially rising to 10 million deaths annually by 2050 [5, 6]. Although a naturally occurring process, the emergence and spread of antimicrobial resistance is accelerated by the use of antimicrobials. Obtaining antimicrobials over-the-counter (OTC) and self-medication can contribute to this process, yet potentially offer little therapeutic benefit, particularly in the absence of appropriate diagnostics [7–9].

In the early stages of the pandemic, diverse therapies were used in response to the rapidly rising mortality [10] and several studies have highlighted the extensive use of antimicrobials [11]. Viral in nature, COVID-19 infections are untreatable by antibiotics, but complications include pneumonia, chronic obstructive pulmonary disease (COPD), mucormycosis and other nosocomial infections, which are driven by bacterial and fungal pathogens and require case management with antibiotics and/or antifungals [10].

Globally, and in Nepal, health systems suffered enormous impacts that inevitably affected health service delivery [12, 13]. With strict lockdown and disrupted healthcare systems, OTC acquisition of drugs and self-medication is thought to have increased during the pandemic [4, 14]. People with perceived COVID-19 and/or those who feared infection resorted to various measures and substances, including traditional and OTC medicines [15]. This was particularly pronounced in community settings in low- and middle-income countries (LMICs), where antimicrobials are often used without a laboratory diagnosis with an antimicrobial susceptibility assay [16–19]. Moreover, LMICs are especially vulnerable to the added burdens of AMR, owing to the constraints in resources, poor disaster preparedness, poor governance, and lack of effective regulatory and legislative bodies [20].

With efforts and resources diverted to the control and management of COVID-19 infections, the pandemic's potential impact on the use of antimicrobials and potential rates of AMR has often been overlooked [21]. Despite its potentially critical impact, few studies have sought to understand how the COVID-19 pandemic affected the use of antimicrobials. Previous studies from Nepal, found that around 70–98% of COVID patients were treated with antibiotics in the hospital [22–24]. Another study found more than half of patients admitted in intensive care unit (ICU) in a tertiary care hospital were prescribed empirical antimicrobials [25]. The use of antibiotics was widespread even when only around half of the COVID infections had bacterial co-infections, and among severe and moderate COVID patients, around 20–25% had antimicrobial resistant bacterial infections [26]. Nonetheless, no studies have used qualitative research methods to examine the underlying social, cultural and health system contributors of antimicrobial use during this period. Drawing on semi-structured interviews with diverse stakeholders, including patients, clinicians and drug dispensers, this article examines the use of antimicrobials, during the pandemic in Kathmandu, Nepal.

## Materials and methods

### Study design

An exploratory cross-sectional study was conducted with three broad stakeholder groups. The study drew on in-depth SSIs with clinicians who have treated COVID-19 patients, drug

dispensers and COVID-19 patients. Patients who were confirmed to have been diagnosed with COVID-19 based on the national COVID management guidelines were considered as respondents in this study. COVID-19 guidelines confirmed the diagnosis based on the polymerase chain reaction (PCR) at test allocated centers. Reflecting on their experience with the COVID-19 event, the study followed broadly a phenomenological theoretical approach. The study followed a COREQ (Consolidated criteria for Reporting Qualitative research) guideline (S1 File).

### Study settings

This study was conducted in and around three major tertiary hospitals in Kathmandu, Nepal. Kathmandu is also most populated urban area and experienced the highest COVID-19 case load and fatalities during the pandemic [27]. The three hospitals were purposively selected for this study that included Sukraraj Tropical and Infectious Disease Hospital (only hospital in the country that is dedicated for infectious diseases), Bir Hospital (the oldest tertiary hospital in the country), and Norvic International Hospital.

Around these hospitals, there were a variety of private healthcare providers, including dispensaries, which were small privately-owned pharmacies with some offering clinical services and were responsible for wide coverage of healthcare services. Although some dispensaries arranged appointments for patients with clinicians, the majority functioned as drug outlets.

### Sampling procedure, recruitment and semi-structured interviews

This study employed the purposive sampling method to select respondents. Participants were selected from the chosen hospitals for this study. At first, a clinician from each of the hospitals was approached and was asked for their suggestions for the patients. Similarly, dispensaries around the hospital and community pharmacies away from the tertiary facilities were approached for the diversity of respondents. Clinicians were recruited based on their exposure to and interactions with COVID-19 patients. Pharmacists or drug dispensers (also known as drug sellers or salespersons) who had qualifications to sell drugs at the pharmacy were selected in the study. Potential COVID-19 patients were recruited based on the history of being COVID-19 positive and seeking treatment at any point during the pandemic. A total of 5–10 participants were targeted for each category to allow the flexibility to adjust the number of participants until data saturation. The number of respondents per each category was deemed adequate based on the content of the interviews where no new information was found after additional interviews [28]. Further informal discussions were held among the additional respondents who echoed the findings.

### Data collection procedures

Two trained research assistants (BD and UTS; postgraduate Nepali researchers in Microbiology with skills in qualitative methods) conducted the SSIs in a quiet location chosen by respondents. SSIs were conducted based on the thematic guide designed by the study authors (S2 File). The investigators BA (MD, DPhil) and KRR (MSc., PhD) trained the research assistants (BD and UTS) and initiated and supervised the procedures (such as obtaining of first few informed consents and execution of SSIs) on their physical presence. After an initial SSI of one person each from the desired stakeholder groups (clinicians, dispensers and patients), investigators (BA and KRR) held a reflection meeting to further instruct and train the assistants. The SSIs had the varying lengths of 13–42 minutes and were conducted in Nepali language.

The SSIs were conducted using a topic/subject guide tailored to each respondent group. The guides were developed by the research team and focused on practices and frequency of prescription, sale and uses of antibiotics for COVID-19 patients, use of OTC antimicrobials

and self- medication, basic awareness of AMR, and how the pandemic has impacted on the use of antibiotics and AMR. For interviews, all respondents were asked for a quiet location, without company. Respondents were interviewed at their workplace in a quiet location often at the convenient locations recommended by them. All interviews were conducted face-to-face and were audio-recorded after obtaining the written informed consent forms. No interviews were repeated, and no participants were consulted for their transcripts.

## Data analysis

Thematic analysis of the transcripts was conducted at various points of the study that entailed discussions of the findings and reflection among the interviewer (BD) and researchers (BA, KRR and SA). These core investigators are from Nepal and have living experience on OTC use of antimicrobials and have bearing on the interpretation of the data. The transcripts and the findings were not shared among the respondents for their opinion, although some of the preliminary findings were re-discussed among the investigators and were prioritized in the subsequent interviews. Once the transcripts were ready, two researchers (BD and BA) independently read line-by-line and coded transcripts in QSR NVivo. Thematic analysis involved using a deductive approach first to categorize the data based on the interview guides which were followed by addition of codes based on the line-by-line readings of the transcripts. All themes, both major and minor were discussed first among the data analysts (BA and BD) and then among the research team. Both major and minor themes identified from this study are the basis of the results in this study. Broadly, following previous qualitative work, data were triangulated from each of the respondents and are presented under the themes. Excerpts were chosen based on their relevance to the themes in addition to the recurrences and uniqueness.

## Ethics approval

The study obtained ethical approval from Oxford Tropical Research Ethics Committee (OxTREC #502–22) and Nepal Health Research Council (Ref#156/2022). Additional information regarding the ethical, cultural, and scientific considerations specific to inclusivity in global research is included in the S3 File.

## Results

### Overview of findings

A total of 30 respondents participated in this study that included 10 each from clinicians, drug dispensers and patients and took place between October 2022 and February 2023 (**Table 1**). Five dispensers among the approached population refused to take part in the study, only two of them expressed their reasons for refusal to participate, and the main reason was that they suspected the interviewer could be an inspection authority or a journalist. Two clinicians could not participate because of their busy schedules and one patient was taking part in SSIs but had to leave for his home during the middle of the conversation. Most respondents from all three stakeholder groups described increased use of OTC medications during the pandemic. Many patients had limited comprehension of antibiotics and AMR and only a few dispensers understood the concept and mechanisms of AMR. Respondents perceived antibiotics as able to reduce the morbidities and mortalities associated with COVID-19. Indeed, patients perceived antibiotics as useful tools to tackle wide varieties of illnesses including the respiratory conditions that occurred during COVID-19. The fear and uncertainty attached to COVID-19 and the potential stigma if and when diagnosed made patients hesitant to visit formal health care services and instead resorted to informal health services. Hesitancy to visit the

**Table 1. Socio-demographic characteristics of participants in the study.**

| SN | Category of respondents | Gender | Age | Education |
|---|---|---|---|---|
| 1 | Dispenser | M | 34 | PCL |
| 2 | Dispenser | F | 23 | PCL/BA Running |
| 3 | Dispenser | M | 45 | PCL |
| 4 | Dispenser | M | 28 | PCL Pharmacy/HA/Masters in Business studies |
| 5 | Dispenser | F | 26 | PCL/B. Pharm. running |
| 6 | Dispenser | F | 44 | PCL Pharmacy/ANM |
| 7 | Dispenser | M | 48 | PCL in Pharmacy /Masters |
| 8 | Dispenser | M | 28 | PCL |
| 9 | Dispenser | F | 34 | PCL/ Bachelor |
| 10 | Dispenser | M | 32 | PCL/ MEd |
| 1 | Clinician | F | 32 | MBBS |
| 2 | Clinician | M | 40 | DM |
| 3 | Clinician | M | 32 | MBBS |
| 4 | Clinician | F | 28 | MBBS |
| 5 | Clinician | M | 34 | MD/Resident Running |
| 6 | Clinician | M | 39 | DM |
| 7 | Clinician | M | 41 | MD |
| 8 | Clinician | M | 31 | MD resident running |
| 9 | Clinician | M | 37 | MD |
| 10 | Clinician | M | 33 | MBBS |
| 1 | Patient | F | 35 | Bachelor |
| 2 | Patient | F | 30 | Literate |
| 3 | Patient | M | 35 | Masters |
| 4 | Patient | M | 23 | Class XI |
| 5 | Patient | M | 28 | Bachelor |
| 6 | Patient | M | 37 | Masters |
| 7 | Patient | M | 25 | Bachelor |
| 8 | Patient | M | 27 | Masters |
| 9 | Patient | M | 34 | Bachelor |
| 10 | Patient | M | 49 | Literate |

**Note:** further socio-demographic characteristics of participants are concealed for anonymity.

formal health care services also made patients to explore home remedies including wide spectrum of untested and unverified herbal products (**Fig 1**).

## Treatment seeking and consumption of antibiotics during the pandemic

Patients shared their experience that they felt general weakness and discomforts even weeks or months after recovery from COVID-19. Respondents experienced severe symptoms whereas others had milder symptoms. Four patients in this study had been admitted to high-dependency care units (HDUs) and/or intensive care units (ICU). Participants had suspected themselves of having been infected with COVID-19 infections even before they were confirmed by tests and sought all kinds of medical assistance.

Most of the patients followed home remedies with some traditional medicines (such as turmeric water, basil water, ginger water) for the first few days of symptoms onset. As the disease progressed, they sought assistance with OTC drugs such as paracetamol, multivitamins and antibiotics. Some patients were admitted to hospital just after three days of disease onset and

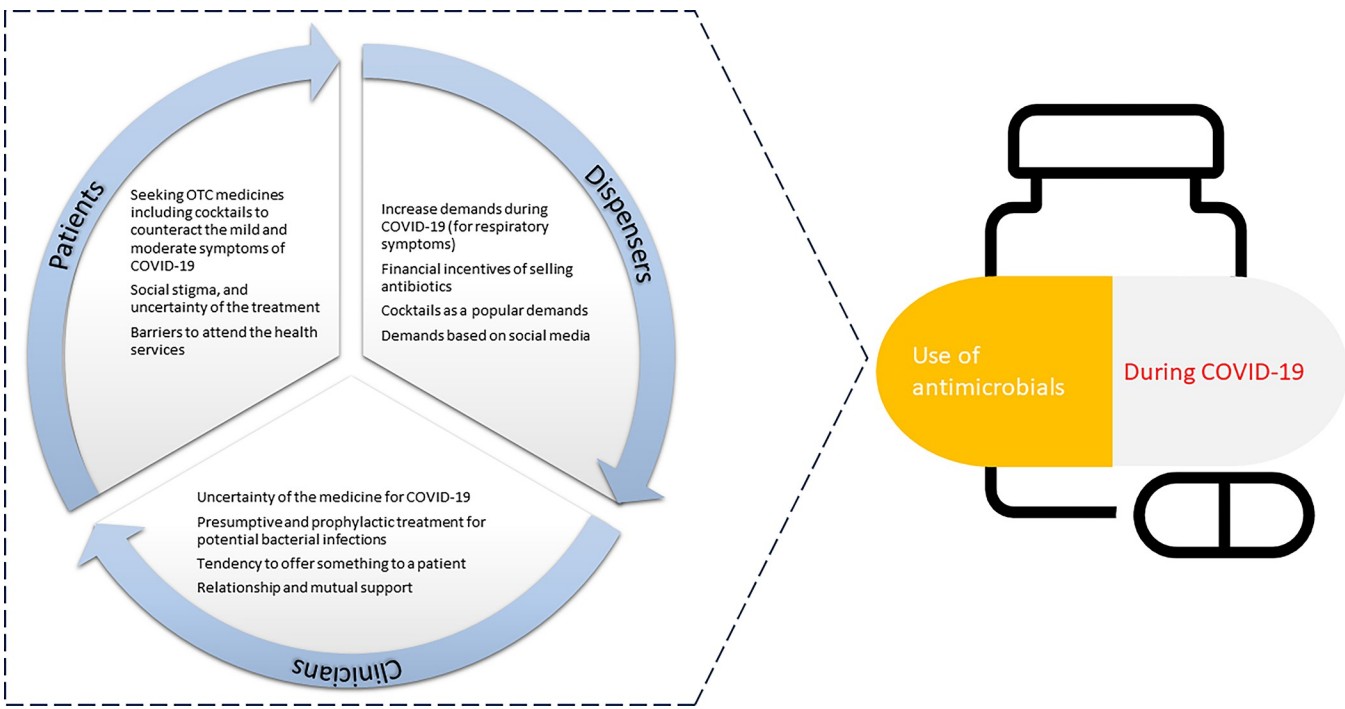

**Fig 1. Thematic findings illustrating contributions to use of antimicrobials during COVID-19.**

had to be treated with oxygen support. Respondents visited laboratories mostly when they suspected that the illness could be COVID-19 and could be transmitted to their family members. Patients also visited laboratories when they failed to resolve their conditions after self-medication, including a wide range of home remedies. Patients shared that most of their suspected close-contacts and acquaintances wanted to evade the laboratory diagnosis of the COVID-19 because of fear of mandatory isolation, social stigma (especially during the first wave in which people used to think of the novel virus as a mysterious agent of the disease), and lack of definitive treatment. Patients explained that even when they were diagnosed as COVID-19, they would not have any treatment available which discouraged them to diagnose their illness. The lack of definitive treatment (with or without diagnosis) also led patients to try combinations of home remedies and OTC medicines that included antibiotics.

Most patients had little understanding of antibiotics, although they had heard about them and some perceived them as strong medicines. Many patients recalled the drug Azithromycin which they had used as OTC medicine during the pandemic, although most did not know if this was an antibiotic. Patients were unable to recall the names of the antibiotics yet many confirmed that Azithromycin was one of the most bought medicines during the pandemic. Other antibiotics were also used (prescribed by dispensers and doctors) but patients could not recall their name.

Many patients had a habit of buying antibiotics OTC around two to three times a year for themselves and family members,. Patients thought that antibiotics were useful, often lifesaving based on their understanding of the term 'antibiotics'. However, some respondents said that they would never buy OTC drugs because of unfavorable experiences during a COVID infection. The utility of antibiotics was reported to be high for the patients with severe symptoms whereas there was little or no use for those with milder symptoms according to clinicians. There was a mixed reaction to the overall use of antibiotics before and after the pandemic,

whereas most of them agreed that there was increased use of antibiotics during the peaks of pandemic. When probed on possible implications of over-use of antimicrobials, patients thought that AMR might have increased due to the pandemic.

Participants also linked using OTC drugs to their hesitancy to visit large private health care institutions because these institutions were deemed to be commercially motivated, which essentially meant that participants were prescribed various diagnostics and medications, sometimes unnecessarily. The diagnostics and number of medications were perceived to be a burden to patients, and thus resorted to evading formal health services by visiting the drug shops. Nonetheless, buying OTC medication was realized to be sub-optimal for their care.

*There could be both benefits and harms. Benefit from the perspective of short-term relief but in the long-term the OTC medicines may bring out several health hazards. Nowadays doctors are waiting patients so as to prescribe and sell more and more drugs to increase their business volumes. This sector has been commercialized by not only pharmacies but also by doctors themselves. Even if a patient does not need antibiotics, they want to prescribe and sell their desired products. Doctors and healthcare facilities today are very far from those who do not have a good income. I could not find any hospital for admission during the pandemic time although I made phone calls to several hospitals in the valley.*

**37 years old, male, patient**

Patients also described their pre-existing habits of seeking OTC medications for common symptoms, such as fever, cold, tonsilitis, sore throats and wounds. This included purchasing an entire dose of antibiotics at once and rarely completed the dose of purchased antibiotics, stopping when symptoms subside, and rarely visiting pharmacies or hospitals for follow up. Most patients revealed that they abandoned the dose after the first three days of taking the drug.

*I buy the entire dosage at once. I rarely go for follow-up; if I am feeling better, why should I?*

**30 years old, female, patient**

## Selling antibiotics

When asked about the notable changes that dispensers had observed during their service-careers, most of them appreciated the advancement in healthcare facilities, improvement in accessibility and raised awareness towards individual and family health. At the same time, dispensers were worried about the progressive ineffectiveness of antibiotics (and other drugs). Such progressive ineffectiveness of antibiotics was attributed to self-medication and patients' demands for specific medications.

*There is a drastic change. During the initial days of my service, a low-dose simple combination used to be sufficient for most of the complaints, which is of no use these days. Patients used to follow our guidelines and suggestions. Nowadays, people [either] are well-educated or pretend to be learned. I do not know, but it is difficult to convince them [these days]. A sense of distrust towards healthcare service providers is well observed in present days.*

**44 years old, female, dispenser**

When discussed about the drivers of OTC medication, the major source of information were various websites surfed through Google and social media.

*They [customers/visitors/patients] are following close-contacts, social media, YouTube and Google search medium rather than specialists. They have more belief in those media than us. I do not say these media are not good, but how would they identify the true information when everything is flooded there? Referring to the examples of someone who has recovered from any specific symptom with a specific drug and exerting pressure for the same drug when similar symptoms are developed has become a new normal these days.*

**45 years old, male, dispenser**

Dispensers were also concerned about the growing distrust in healthcare workers and the government. Distrust towards the dispensers were attributed to perceptions of healthcare as motivated by monetary incentives rather than the wellbeing of the patients. Such distrust was thought to be the main drivers of seeking OTC medications, which apparently helped patients bypass the formal health care workers and the service institutions.

Dispensers agreed that they had prescribed antibiotics mostly based on their presumptive diagnosis rather than the diagnosis based on the lab confirmation. During the pandemic, dispensers reported an increase in the sales of antipyretics, analgesics, multivitamins, and antibiotics as major drugs to mitigate the symptoms of COVID. There were mixed reactions on the overall sales of antibiotics during the pandemic. Dispensers also reported that there were no significant increase in number of visitors during the first wave of the pandemic but there was a substantial increase during second and third waves.

Nonetheless, dispensers agreed that they had received increased requests from patients for antibiotics during COVID pandemic compared to pre-COVID times. Related to the type of patients in terms of demanding the antibiotics, dispensers shared that the seemingly educated ones generally dictated the demands while illiterate patients generally came with previous drug leaflets and empty blister package to demand for exact same medications, most of which were antibiotics. On some occasions, patients demanded antibiotics, referring to the need for their family, friends and relatives.

Based on the dispensers' opinion, respiratory and diarrheal illnesses were the major reasons for sales of antibiotics. Driven by excessive pressure from patients and self-prescription (medication), dispensers did not reject selling those drugs sought by patients although they tried convincing patients not to opt for antibiotics at the first step.

*We would try to figure out the person psychologically. If he seems to be convincing, of course we would. But if the person seems to be stubborn and even on the verge of switching my pharmacy [in case he does not get his desired drugs], we simply deliver our intended message in not so stressing manner. After all, it is our business too. Who would like to lose a valued customer?*

**44 years old, female, dispenser**

The respondent further added: *People are mostly preoccupied. Even if the handwriting of the doctor is not well understood and even if there were availability of quality drugs from another company, people would not like to listen. They hear only what they want to hear. They [would] rather visit a dozen pharmacies for their drug of choice but would hardly get convinced.*

Apart from excessive pressures from patients, non-standard doses were of another concern. Some patients used to ask for just a couple of tablets of antibiotics and denying such requests was out of scope for dispensers as patients would switch to generally an adjacent drug store.

Dispensers from community pharmacies described how the OTC sales of antibiotics comprised more than two-thirds of the total sales of antibiotics. In general dispensers did not have

their lab for confirmatory diagnosis, and thus had to resort to empirical dispensing of antibiotics. The antibiotics most often dispensed included azithromycin, amoxicillin, cefixime, ampicillin, levofloxacin and clavulanic acid during and after the pandemic. Dispensers believed that these antibiotics were the most extensively used drugs in terms of covering most of the infectious diseases that they have encountered so far. Dispensers did not reject selling OTC antibiotics although some dispensers of hospital pharmacies revealed that they recently began rejecting it because of the fear from regulatory bodies.

> *We have recently begun to reject some of the requests. It's mainly because of fear from authorities as there are stringent regulations being implemented these days.*

**26 years old, female, dispenser**

Most of the drug dispensers had no specific knowledge regarding AMR and the public health crisis due to its emergence. When asked about the knowledge and cautionary measures required to counter the COVID-19, respondents were aware of the symptoms, disease manifestations and precautions to be taken while dealing with the infected individuals. OTC drugs were offered to patients based on the symptoms that resembled COVID-19. Regarding the need and effectiveness of antibiotics in the management of COVID-19, the respondents believed that antibiotics had impact on COVID-19 but did not know the specific mechanisms, how it cured the infection. In terms of timing, dispensers believed that patients attended their drug shops once their home remedies failed to cure.

Dispensers also reported that antibiotics were expensive but even so demands for incomplete dosage were problematic–concerns motivated by the potential revenues of selling the full course, but they saw the adversities of underdosing. The excessive use of antibiotics, and the incomplete dosing could contribute to the existing problem of AMR. Irrespective of their own shortcomings, dispensers knew and expressed that OTC use of antibiotics needed scrutiny and wanted to convey message to patients.

> *I request you all the patients to avoid quackery (pretension of well understanding of antibiotics among educated and mass people during these days) and listen to the trained healthcare professional. Further I stress on the need for a completion of a full course of treatment among the patients undergoing antibiotic therapy.*

**48 years old, male, dispenser**

## Practices regarding antibiotics prescription

When asked about the specific changes before and after the pandemic, most clinicians witnessed increased respiratory complaints and other post-COVID illnesses including, psychological problems (anxiety, increased heartbeat, and panic attack) and heart diseases. Clinicians were aware of OTC use of antibiotics, AMR and their potential consequences. A significant majority of the clinicians had experienced pressure to prescribe drugs from patients. An intensivist shared his experience as

> *Most patients ask drugs [basically painkillers and some antibiotics] for themselves complaining of the pains, discomforts or other illnesses. However, we never prescribe unnecessary drugs. We tell the patients that the drugs they intend to consume may not be necessary and useful rather may be harmful. We do not get them convinced.*

**41 years old, male, clinician**

Clinicians listed the following symptoms as the major drivers behind OTC sales of antibiotics: respiratory tract infections (sore throat, common cold, cough, and tonsillitis), fever and urinary tract infections. These symptoms were similar to those of COVID-19 which might have led to increased use of OTC drugs during the pandemic. Resemblance of the COVID-19 symptoms to other respiratory infections also promoted OTC use of antibiotics because of the seemingly familiar symptoms which were generally cured by the OTC drugs in the past. In the second and third waves, the demands on clinicians to prescribe antibiotics increased. Clinicians also reflected on their services to rural regions. The scenario was intense in rural regions where clinicians were under pressure to offer antibiotics because the patients travelled from far to attend the health facilities. These patients were unlikely to follow-up for monitoring of symptoms and took complacence through the antibiotics irrespective of their need and impact on their illnesses. Sometimes clinicians had to prescribe antibiotics to avoid the potential hostility with patients resulting from sending patients empty handed.

*I think it [handling pressure from patients] mainly depends on the physician's side. If one can convince the patients, probably they would likely understand. Some proportion (around 10%) would still be not convinced. If the physician pays attention and spares ample time, the patient would likely be convinced but the problem with us is that we cannot give enough time and counseling [due to the high number of visitors per physician]. And some physicians are taking this reason [default condition] as a means of excuse even when they have enough time. In rural settings, sometimes physicians in far off districts do not want to displease the locals, as physicians have to get familiar with them so that they can continue their job smoothly. Who wants to confront the locals even if they are constantly pressurizing for a specific drug?*

**31 years old, male, clinician**

Paracetamol and pain killers, anti-diarrheal, antibiotics, salbutamol for breathlessness, and other topical antibiotics for wounds and other skin infections were recalled as the major drugs being asked by the patients. Unsurprisingly, most of these drugs were common OTC medications all over the country.

*I can tell you with certainty that asking for antibiotics and OTC use of antibiotics is rampant all over the country. Amoxicillin, ciprofloxacin are the chief used antibiotics in rural settings as it is provided freely by the government and are common medicines for villagers.*

**40 years old, male, clinician**

Some physicians expressed worries regarding the pressure to prescribe drugs they used to receive from well-educated patients. Pressure to prescribe stemmed from the information they accessed over the non-specific web search engines (such as Google) of the internet. Some experienced an increased pressure of drugs in post-COVID time referring to the patients' altered behavior in seeking medical care.

*It has increased in post-COVID time. It may be due to the fact that before pandemic patients could access hospital facilities easily. However, during the pandemic, this easy access was greatly reduced due to lockdown and other travel restrictions which compelled them to rely on pharmacies available nearby. In this way, pharmacies may have prescribed and/or sold unnecessary drugs as a lucrative means which may have altered the patients' behavior in seeking medical care and purchasing drugs.*

**39 years old, male, clinician**

Physicians were prescribing antibiotics daily as an essential aspect of their treatment therapy. Azithromycin, amoxycillin, cefixime, ampicillin, levofloxacin, amoxiclav, cephalosporin, and ofloxacin/ciprofloxacin were frequently prescribed antibiotics by physicians. Participants recognized azithromycin as the most sold antibiotic in the country while some of them listed azithromycin and amoxicillin as comparable drugs in view of business volume.

When asked about the antibiotic prescriptions, clinicians resorted largely to empirical practices referring to the reasons that most of the patients were already taking antibiotics given OTC and visited their clinic when symptoms did not subside. This particular behavior interfered the diagnosis, as culture and drug susceptibility tests were bound to be altered by the OTC use of antibiotics. Thus, clinicians prescribed higher generation antibiotics to counteract the unresolved infections. Clinicians also shared that most patients who visited the clinic came from remote and rural regions and did not have time, money and patience to wait for the culture results which also obliged clinicians to be context friendly by offering empirical antibiotics at an OPD basis. Clinicians also used antibiotics based on the length and severity of the infection.

*Duration and severity of the onset of the symptoms play a crucial role in our response. If the duration is shorter, we don't go directly with antibiotic treatment rather we aim for symptomatic management. However, if there is [an] increase in duration and severity, we have to go for antibiotic therapy. It's again empirical than following any specific protocol.*

**32 years old, female, clinician**

Clinicians had experienced poor follow-up among patients who were prescribed antibiotics. Most patients in their experience did not complete the full course of treatment. Patients also brought the old prescriptions, empty blister packs and made request for the prescription. As most of the clinicians had past experiences of service in the peripheral/rural part of the country, there was a unanimous view that OTC drugs (including antibiotics) were prevalent in any corner of the country. They expressed that '*empirical prescription will not be replaced anytime soon, given the lack of diagnostic facilities in the majority of the healthcare centers that covers about two-third of the total population.*'

Although the physicians believed that prescribing antibiotics were one of the counter strategies to tackle the disease progression (e.g. secondary bacterial infections), they did not consider the therapy at first. Clinicians were well aware that antibiotics had nothing to do with the viral infections including COVID-19. But as the disease progressed, secondary bacterial infections were prevalent among severely ill patients. Antibiotics were therefore inevitably prescribed to prevent secondary bacterial infections. Antibiotics were considered a lifesaving option. Nonetheless, clinicians echoed with both patients and dispensers that self-medication and subsequent pressure for OTC was more prevalent among patients with milder symptoms than the ones who were admitted. Clinicians shared some of the most prescribed antibiotics during the pandemic included azithromycin, piperacillin/tazobactam and cephalosporins among the patients who were admitted at the ICU.

When doctors were asked about the future applicability of antibiotics in COVID-19, they could not rule out the drug's necessity among COVID patients with co-morbidities and other potential bacterial (secondary and nosocomial) infections during the course of infection. Physicians were aware about the drug resistance, and believed that AMR may have been flared up due to excessive and inappropriate use of antibiotics during the pandemic. Clinicians shared serious concerns about the potential consequences that can result from AMR and reflected on their own roles and obligation to combat burgeoning surge of AMR.

*Nowadays, we admit patient in ICUs and request for the antibiotic sensitivity test where we encounter the resistance to almost all of the available drugs. Who is responsible for this? Definitely we are. We physicians are mainly responsible for this situation. We are taught to escalate rather than to deescalate the situation [sarcasm]. We are much poorer when it comes to antibiotic stewardship. We merely follow our seniors rather than questioning the long-held practices. There is over consciousness, lack of knowledge and limitation in our diagnostic capacities [which is] a major problem. But we should take the responsibility from our side rather than blame. And from the policy level side, it is seriously affected by various problems. These are the situations that have prevailed among us.*

**34 years old, male, clinician**

## Discussion

This is the first study in Nepal to examine the use of antibiotics during the COVID-19 pandemic from the experiences of COVID-19 patients, dispensers and clinicians. The use of OTC antibiotics increased during the COVID-19 pandemic for several reasons. The high empirical treatment through prescription from clinicians and OTC-based use of antibiotics echoes studies from Nepal [22–26], Scotland [29], Saudi Arabia [30], Peru [31], and Bangladesh [32]. Indeed, the overuse of antibiotics during the pandemic has been reported globally [11]. Several follow up studies have confirmed the overuse and inappropriate use of antimicrobials among COVID-19 patients [33–40]. COVID-19 was deemed to be a special circumstance that added extraordinary pressure on both physicians and drug dispensers [29]. Although clinicians perceived the need to halt the severe infections and prevent the potential secondary infections following COVID-19, dispensers' incentives for OTC sales of antibiotics were primarily patients' demands. This finding reflects reports from Eastern Mediterranean regions [41] and previous research in Nepal [42–44]. Although patients did agree that they may have visited dispensaries more frequently, a lot of these patients were also unaware of what constituted antibiotics and the meaning and consequences of AMR. COVID-19 created a conducive background for all stakeholders to overuse antibiotics. The use of antibiotics through dispensary outlets remains sub-optimal and detrimental because they were also underdosed and based on the non-expert (patients') demands, and dispensers' incentives.

In line with previous studies [43, 44], respondents' knowledge on antibiotics were poor, though they were seen as a powerful group of drugs. This echoes findings from research in Bangladesh [45]. Although dispensers did have knowledge about antibiotics, they did not understand the concept and mechanisms of AMR. Inevitably, very few patients knew what antibiotics were and AMR was in general, a complex concept.

### OTC use of antibiotics during COVID-19

Most of the patients wanted to avoid the confirmatory diagnosis because of the social stigma attached to COVID-19, specifically, fear of social isolation, mandatory quarantine, and also the lack of definitive treatment for it. Some of these drivers were echoed by previous study from Nepal [46] while a study from 28 countries documented social stigma as the most important factor in contributing the perceived risk and fears as the root cause of averting health-seeking behaviors and practices at formal health services [47–51]. The other reasons to avoid seeking definitive diagnosis and treatment were because patients considered COVID symptoms as simple case of 'common cold' owing to the symptomatic resemblance and resorted to OTC antibiotics.

Patients in this study followed home remedies and OTC medicines before they sought medical attention in hospitals. Our finding has been corroborated by several other studies from Asian and African countries [52–54]. Private pharmacies and dispensers comprised the larger chunk of informal prescription and sales of OTC medicines to these patients; this finding echoes with several previous reports [55–57]. Other reasons behind informal prescriptions were patients' self-medications and induced pressure on dispensers, which were often guided by their close-contact and social media. Similar compelling factors were also outlined in the previous studies [15, 58]. Self-medications with home-based remedies were also widely practiced in Nepal that mostly comprised boiled water with basil leaves, ginger and turmeric powders and such practices are prevalent around the globe [52–54, 59]. The use of home-based self-medications also may have distorted the early treatment seeking behavior in formal services. Self-medication was prevalent among medical students too, more than half of medical students were found to self-medicate using OTC purchase [60]. Most of the patients perceived lowered self-esteem and psychological crises in the course of infection as seen in other studies [48] nonetheless, promoting confidence by recommending wide variety of unproven medications including herbs and unlicensed products can deteriorate the clinical course of the illnesses ultimately affecting treatment seeking behavior [61].

Dispensers also echoed the heightened use of antibiotics during the pandemic. The major reasons for high use of antibiotics were attributed to patients' demands (present also during non-COVID times). Their default position on selling antibiotics was compounded by patients' claims and demands, which they believed were mostly driven by social media and discussions with relatives, neighbors and friends [15, 58, 62]. The uncertainty of the pandemic in terms of what COVID is, how it is caused, and the potential causative agents are known contributors to the OTC increases and follows a global trend [32]. OTC demands for antibiotics were a pre-existing trend and behavior in Nepal [20, 43, 44] and were accentuated by the added complexity of COVID-19. This also included buying incomplete doses and poor adherence to the antibiotics when they needed it [63, 64]. Although dispensers' acceptance of increased OTC sales of antibiotics is clear indication of how pandemic accentuated the inappropriate use of antibiotics, dispensers were restrained by their commercial interests in not losing the customers by complying with the demands [63, 65]. OTC sales of antibiotics during COVID-19 were found high among dispensers when approached by a simulated client [30]. Knowledge that sales of antibiotics without prescriptions and potential adversities was understood to some extent by the dispensers, but the policy restrictions and ethical good practice were beyond their scope when it came to daily practice. During interactions, dispensers also attempted to display their knowledge about the adversities of dispensing antibiotics even if the practice did not approve their opinion.

Clinicians echoed with both patients and dispensers that there was an accentuated use of antibiotics during the pandemic and the pressure on prescribing antibiotics were felt by the clinicians. Two kinds of pressures were explained by the clinicians, one that was intrinsic inherent in their own practice and another one was extrinsic that emerged from patients and their relatives. These findings were also supported by previous studies [29, 66–69]. COVID-19 manifested as respiratory illnesses and there was an increase influx of patients who either already had OTC antibiotics or were brought severely ill at the tertiary care centers. The obvious delays in the diagnosis of the causative agents (such as bacterial infections) meant that clinicians had to resort to empirical treatment to prevent the secondary infections [6, 70]. Also, the increase in respiratory symptoms compelled clinicians to offer antibiotics to prevent its further progression. In addition, clinician's practice to prescribe or offer something to a patient presenting with respiratory illness is also socially woven to be seen as kind and 'giving' rather than being strict on prescribing and adding disappointments to patients [29]. Antibiotic prescriptions for

patients were also a symbolic gesture of social niceties, relationship and trust. Especially, during the pandemic where trust towards health care workers was endangered due to uncertainty, antibiotics may have had an impact as a social binder and a source of trust and relationship—a commonest denominator contributing to the acceptance of health services and interventions [61, 71].

### Implications for redressing OTC use of antimicrobials during pandemic

COVID-19 was a special circumstance that exerted extraordinary pressure on the OTC use of antimicrobials, and future pandemics are likely to reproduce the phenomenon. The first step in mitigating the excess use of antimicrobials can begin from being cognizant of its demands, followed by launching multi-pronged approaches to reduce its demands such as through stakeholder and community engagement [72]. With pandemics established as a driver of AMR, pandemic preparedness programs should include strategies to reduce the use of antimicrobials [73]. Revised policies to counteract excess and inappropriate use of antimicrobials during pandemic should be incorporated in contingency plans. Finally, a robust surveillance system should be employed to monitor the use of antimicrobials and antimicrobial resistance which should also include regular audits of both prescription-based sales and OTC-based sales of antimicrobials [74] without disruption during pandemic.

### Strengths and limitations

Respondents (mostly dispensers and clinicians) in this study may have been affected by the social desirability bias. Thus, their views and perspectives might be different from their practice, nonetheless, follow-up and indirect questions were asked to report the overall behavior of stakeholders that may have offered space to discuss the topics more openly without self-attributing. Some of the respondents, especially dispensers, tried to avoid the interviewer because of the qualms and suspicion that they may have been monitored or inspected by authorities or journalists. Although interviewer attempted to balance the gender among respondents, higher proportion of clinicians were male reflecting the sex ratio among the doctors' population in Nepal. Future studies could take a more quantitative approach to the use of antimicrobials during pandemic with larger samples including the factors affecting its use and potential mitigation strategies. Owing to the high OTC use of antimicrobials during pandemic in Nepal attributable to the poor access and quality of formal health services, future studies should explore on ethical dimensions of policies regulating the use of antimicrobials. In addition, exploring use of antimicrobials in animal and agricultural products requires a closer examination using an ethical lens.

### Conclusions

The COVID-19 pandemic strongly influenced the use of antimicrobials–with or without a prescription–in Nepal. The pandemic placed extraordinary pressure and demands on physicians and drug dispensers. While clinicians prescribed antibiotics to arrest the severity of infection, prevent secondary infection and to maintain the patient's demands, dispensers' selling of antibiotics was mostly motivated by their commercial interests, and prompted by patients' demands. The increase in OTC sales of antibiotics was also inappropriate because of the sale of incomplete dosage/duration. Regulatory policies on antibiotics need a revised contingency plan specifically to tailor for special circumstances such as outbreaks or epidemics of infectious (respiratory) diseases such as COVID-19.

## Supporting information

**S1 File. COREQ (Consolidated criteria for Reporting Qualitative research) guideline.**
(PDF)

**S2 File. SSI thematic guide for dispensers, patients and clinicians.**
(PDF)

**S3 File. Inclusivity.**
(PDF)

## Acknowledgments

We are grateful to all the participants who generously participated in this study.

## Author Contributions

**Conceptualization:** Binod Dhungel, Sunil Pokharel, Abhilasha Karkey, Direk Limmathurotsakul, Phaik Yeong Cheah, Christopher Pell, Bipin Adhikari.

**Data curation:** Binod Dhungel, Bipin Adhikari.

**Formal analysis:** Binod Dhungel, Bipin Adhikari.

**Funding acquisition:** Bipin Adhikari.

**Investigation:** Binod Dhungel, Upendra Thapa Shrestha, Alisha Bhattarai.

**Methodology:** Binod Dhungel, Bipin Adhikari.

**Project administration:** Sanjib Adhikari, Nabaraj Adhikari, Komal Raj Rijal.

**Resources:** Binod Dhungel, Bipin Adhikari.

**Software:** Binod Dhungel, Bipin Adhikari.

**Supervision:** Prakash Ghimire, Komal Raj Rijal, Phaik Yeong Cheah, Christopher Pell, Bipin Adhikari.

**Validation:** Binod Dhungel, Bipin Adhikari.

**Visualization:** Bipin Adhikari.

**Writing – original draft:** Binod Dhungel, Bipin Adhikari.

**Writing – review & editing:** Binod Dhungel, Upendra Thapa Shrestha, Sanjib Adhikari, Nabaraj Adhikari, Alisha Bhattarai, Sunil Pokharel, Abhilasha Karkey, Direk Limmathurotsakul, Prakash Ghimire, Komal Raj Rijal, Phaik Yeong Cheah, Christopher Pell, Bipin Adhikari.

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
