## [Decision Letter · Decision Letter 0]

22 Sep 2023

PGPH-D-23-01183

Use of antimicrobials during the COVID-19 pandemic: a qualitative study among stakeholders in Nepal

Dear Dr Bipin Adhikari,

Thank you for submitting your manuscript to PLOS Global Public Health. After careful consideration, we feel that it has merit but does not fully meet PLOS Global Public Health’s publication criteria as it currently stands. Therefore, we invite you to submit a revised version of the manuscript that addresses the points raised during the review process.

**Introduction**

Lines 78-84 – The authors have briefly mentioned what other studies in this particular area have done. Perhaps this could be expanded a little more to explicitly identify the gaps in knowledge surrounding this topic and why this study has been conducted? How does this study help to fill that gap?

**Methodology**

Line 117-126 – Is there a specific reason why 30 participants were selected?

**Results**

Line 108 – was asked instead of were asked

Line 212 – recall their names instead of name

Line 168 – A lot of the patients did not have knowledge about what constituted antibiotics and AMR

& Line 220-221 – Most of them thought that AMR might have increased due to pandemic

The two statements seem contradictory. How can the patients not have knowledge of AMR but think that it might have increased due to the pandemic?

Line 287 – repetition of ‘they’

Line 302 – two-thirds instead of two-third of total sales of antibiotics volume

Line 325 – scrutinization is not a word, perhaps ‘scrutiny’ would be appropriate

Line 347-348 – Second and third waves were deemed to mount pressure among clinicians as well to prescribe antibiotics

This needs to be rephrased.

Line 348-350 – The scenario was intense in rural regions where clinicians were under pressure to offer antibiotics because the patients travelled from far to attend the health facilities.

This study was conducted in an urban area. Was the scenario in rural areas mentioned by a clinician? If not a citation is needed.

**Discussion**

Line 440 – OTC-based instead of OTC based

Line 469-484 – patients’ practices of using home remedies provide an interesting insight into their thought processes and health seeking behaviour during the pandemic. However, such a detailed account might not be needed under the sub-title “OTC use of antibiotics during Covid-19”. Mention couple of sentences but an entire paragraph detailing what constitute for home remedies is not needed.

Line 483 – unevidenced is not a word

The authors also need to explain the implications of patients’, dispensers’ and clinicians’ practices might have on Nepal’s AMR crisis.

Regarding patients experience, the discussion briefly mentions self-medications, pressuring dispensers etc. but mainly focuses on home remedies. But the study’s focus is on antibiotics and AMR, so the authors need to expand bit more on those topics. For example, what are the implications of over-usage/over-prescription of antibiotics mentioned by the dispensers and clinicians?

Do the authors have suggestions for future research directions?

We look forward to receiving your revised manuscript.

Kind regards,

Dinesh Bhandari, Ph.D.

Academic Editor

Journal Requirements:

2. Please include a complete copy of PLOS’ questionnaire on inclusivity in global research in your revised manuscript. Our policy for research in this area aims to improve transparency in the reporting of research performed outside of researchers’ own country or community. The policy applies to researchers who have travelled to a different country to conduct research, research with Indigenous populations or their lands, and research on cultural artefacts. The questionnaire can also be requested at the journal’s discretion for any other submissions, even if these conditions are not met.  Please find more information on the policy and a link to download a blank copy of the questionnaire here: https://journals.plos.org/globalpublichealth/s/best-practices-in-research-reporting. Please upload a completed version of your questionnaire as Supporting Information when you resubmit your manuscript.

3. Please amend your detailed Financial Disclosure statement. This is published with the article. It must therefore be completed in full sentences and contain the exact wording you wish to be published.

b. If any authors received a salary from any of your funders, please state which authors and which funders.

4. Please provide separate figure files in .tif or .eps format only and remove any figures embedded in your manuscript file. Please also ensure all files are under our size limit of 10MB.

Additional Editor Comments (if provided):

Reviewers' comments:

Reviewer's Responses to Questions

**Comments to the Author**

1. Does this manuscript meet PLOS Global Public Health’s publication criteria? Is the manuscript technically sound, and do the data support the conclusions? The manuscript must describe methodologically and ethically rigorous research with conclusions that are appropriately drawn based on the data presented.

Reviewer #1: Partly

Reviewer #2: Yes

2. Has the statistical analysis been performed appropriately and rigorously?

Reviewer #1: N/A

Reviewer #2: N/A

3. Have the authors made all data underlying the findings in their manuscript fully available (please refer to the Data Availability Statement at the start of the manuscript PDF file)?

Reviewer #1: Yes

Reviewer #2: Yes

4. Is the manuscript presented in an intelligible fashion and written in standard English?

Reviewer #1: Yes

Reviewer #2: Yes

5. Review Comments to the Author

Reviewer #1: The authors did a good job explaining the factors that affected usage and prescription of antibiotics among patients, pharmacies, and clinicians in Nepal. In general, the study is important as it highlights some of the issues of global health importance like antimicrobial resistance and the need to understand the underlying drivers among people that exacerbates the current situation. I have included here comments that the authors can use to improve their manuscript.

Reviewer #2: Discussion is not sufficient so the authors are requested to add more literature on discussion.

Authors are requested to explain the member checking process if they have done during this study and their reflexivity as well.

6. PLOS authors have the option to publish the peer review history of their article (what does this mean?). If published, this will include your full peer review and any attached files.

**Do you want your identity to be public for this peer review?** For information about this choice, including consent withdrawal, please see our Privacy Policy.

Reviewer #1: No

Reviewer #2: No

---

## [Editor Report · Decision Letter 1]

9 Oct 2023

Use of antimicrobials during the COVID-19 pandemic: a qualitative study among stakeholders in Nepal

PGPH-D-23-01183R1

Dear Dr Adhikari,

We are pleased to inform you that your manuscript 'Use of antimicrobials during the COVID-19 pandemic: a qualitative study among stakeholders in Nepal' has been provisionally accepted for publication in PLOS Global Public Health.

Best regards,

Dinesh Bhandari, Ph.D.

Academic Editor